# FoGE: Fock Space inspired encoding for graph prompting

**Sotirios Panagiotis Chytas**[1]    **Rudrasis Chakraborty**[2]    **Vikas Singh**[1]

[1]University of Wisconsin-Madison    [2]Lawrence Livermore National Lab
chytas@wisc.edu, vsingh@biostat.wisc.edu    rudrasischa@gmail.com

## Abstract

Recent results show that modern Large Language Models (LLM) are capable of understanding and answering questions about structured data such as graphs. Existing proposals often use some description of the graph to create an "augmented" prompt fed to the LLM. For a chosen class of graphs, if a well-tailored graph encoder is deployed to play together with a pre-trained LLM, the model can answer graph-related questions well. Current solutions to graph-based prompts range from graph serialization to graph transformers. In this work, we show that the use of a parameter-free graph encoder based on Fock space representations, a concept borrowed from physics, is remarkably versatile in this problem setting. The simple construction, with a few small adjustments, can provide rich and informative graph encodings, for a wide range of different graphs. We investigate the use of this idea for prefix-tuned prompts leveraging the capabilities of a pre-trained, frozen LLM. The modifications lead to a model that can answer graph-related questions – from simple graphs to proteins to hypergraphs – effectively and with minimal, if any, adjustments to the architecture. Our work significantly simplifies existing solutions and generalizes well to multiple different graph-based structures effortlessly.

## 1 Introduction

Large Language Models (LLMs) excel at tasks like question answering, sentence completion, translation, and even solving undergraduate-level math problems [1, 2]. However, they sometimes need additional data unavailable during training. For instance, a model trained on data up to a specific date may struggle with the ever-changing news cycle [3, 4]. To prevent responses from becoming outdated, or to integrate non-public/proprietary data and domain-specific terminology, models need extra context. Retrieval Augmented Generation (RAG) describes this process of retrieving and integrating extra information to an LLM during its generation process. While multiple different approaches have been proposed for the retrieval piece, a common solution for integrating additional information is In-Context Learning (ICL) [5, 6, 7, 8, 9]. ICL allows additional information to be included with a prompt, guiding the model to generate responses aligned with the extra context. This method is useful as it does not require retraining the LLM and can be applied to proprietary models like GPT [10] by adding a text description of the extra information.

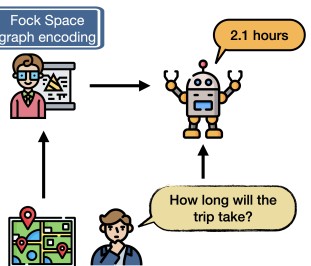

Figure 1: Augmenting LLM's capabilities by prompting them with carefully encoded graphs.

ICL-type ideas are also being studied for utilizing not just additional/new data but also novel input formats/modalities, such as tables and graphs [11, 12, 13, 14]. While specialized models will still perform better at specific tasks, LLMs can serve as general-purpose reasoning machines, capable of answering questions about the provided modality beyond the training labels. Several recent results

39th Conference on Neural Information Processing Systems (NeurIPS 2025).

have reported success at "serializing" such structured data-types into a text-form description that can be easily used within ICL. For tables, the serialization is not too complicated [11, 12], but more care is needed for graphs. While different types of graphs can all be handled by the same pipeline, the efficacy of the model varies [15, 13, 14]. Further, it has been observed that specific design choices to "textify" the graph can influence performance and additionally, prompting techniques can have more than a small impact on results [15]. What will work well in a specific setting depends on both the question at hand as well as the characteristics of the data [16, 17].

**Prefix-tuning.** One option to address the issues above is "prefix-tuning" [18]. A specialized graph encoder translates the underlying graph into embeddings that can be fed directly to an LLM, removing the need for a text description. Although not training-free, the LLM remains *frozen*, and only the *relatively smaller* graph encoder is trained. This approach works well, often surpassing ICL-based methods [19, 20, 21]. However, using a specialized graph encoder can be challenging due to the *variety* of graph types, and multiple works have proposed modifications of GNNs that suit their needs. For example, GraphToken [16] can encode only simple graphs, while GNP [22] constructs a complex pipeline to handle large graphs and extract subgraphs. GraphLLM [17] combines a transformer and a GNN (about 100M parameters), requiring detailed text descriptions for each node. Adapting these models to different graph types (e.g., protein-derived graphs or hypergraphs) is difficult; even familiar graph types need adjustments for new tasks.

**Context of this paper.** ICL-based approaches for graphs primarily involve converting graphs to text, while prefix-tuning with graphs uses modules to extract richer, *task-relevant* structures, requiring larger sample sizes and more compute. A key question is whether we can achieve powerful, task-agnostic graph representations that are as easy to obtain as ICL-based methods. Could a lightweight adapter map these rich (but task-independent) representations into the LLM embedding space, making prefix-tuning effective for various tasks? Recent results hint that this may be viable [23]. For instance, a *single linear layer* can transform an arbitrary image encoder's outputs to align with CLIP's text encoder embeddings [24]. If our graph encoding captures the graph's information and structure well enough, a similar adapter could work with a pre-trained LLM to offer good performance. This approach's success depends on the quality of the graph representations. We ensure this by invoking a mature concept from mathematical physics, called Fock Spaces [25], whose practical instantiation yields almost lossless task-agnostic graph embeddings. Our findings show that a linear adapter with these representations yields competitive performance, handling complex graph questions and diverse structures like hypergraphs and proteins. The **main contribution** of this paper is the Fock-space inspired encoding of diverse graph-based structures, ranging from simple graphs to those obtained from proteins. We provide code for grounding LLMs using our graph encodings as prompts and profile the performance of this pipeline relative to baselines, on diverse datasets.

## 2 Deriving Fock space based Graph Representations

We will first review notations/results which will together provide the conceptual pipeline. While graphs serve as representative examples here, the rationale for structured data such as tables is similar.

**Setup/rationale.** Consider a graph $G = (V, E)$ with a vertex set $V$ and an edge set $E$; $|\cdot|$ denotes set cardinality. We define the *incidence matrix* [26], $I$ to be of shape $|V| \times |E|$ where $I_{ij} = 1$ if *edge $j$ ends at vertex $i$*, $-1$ if *edge $j$ starts at vertex $i$* and 0 elsewhere. Let $|V| = n$. It is common to represent graphs via graph spectra derived from the Laplacian's eigenvalues. This is effective for studying global properties of graphs like connectivity/symmetries [27, 28] but less so for capturing localized relationships between individual entities (nodes, edges, faces) within the graph. It turns out that an interesting direction using Clifford Algebra, shown to be effective in geometric problems

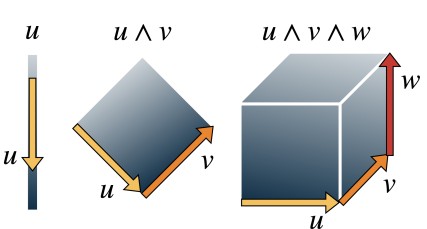

Figure 2: Single, Bi- and Tri-vectors in Clifford Algebra with wedge products.

in machine learning [29, 30, 29, 31, 32], provides us tools for representing various graph elements (nodes, edges, faces) in a nice algebraic structure [33] **at once**. *Why?* Graphs can be embedded and manipulated in a geometric space [34], and in principle, their spectral properties can also be studied. We briefly summarize the concept to assess its benefits and challenges.

## 2.1 Clifford Algebra and Graph Representations

**Clifford Algebra.** We start with a vector space $W$ over a field $K$ (e.g., $\mathbb{R}$ or $\mathbb{C}$). Vectors in $W$ support operations like addition, scalar multiplication, and subtraction. We also equip $W$ with an inner product $\langle \cdot, \cdot \rangle$ that measures relationships between vectors. On top of our vector space $W$, we construct the tensor algebra $T(W)$. We can think of $T(W)$ as containing all possible ways to multiply vectors together using the tensor product: it includes the original vectors from $W$, all pairs of vectors, all triplets, and so on. It also includes sums and scalar multiples of these products. For creating the Clifford Algebra, the main step is to modify this tensor algebra by imposing a specific constraint. We define an ideal $I(W)$—essentially a special subset that acts as a "filter"—generated by expressions of the form $w \otimes w + \langle w, w \rangle \mathbf{1}$. Here, $w \otimes w$ is a vector multiplied (via tensor product) by itself, and $\mathbf{1}$ is the multiplicative identity. This ideal has a closure property: multiplying any element from $I(W)$ with any element from $T(W)$ yields another element in $I(W)$. This constraint will impose the algebraic relations (particularly anticommutation) we need for representing graph structures.

**Definition 2.1.** Let $W$ be a vector space over a field $K$, equipped with a quadratic form $q : W \to K$. The **Clifford algebra** of $(W, q)$, denoted $\mathfrak{Cl}(W, q)$, is the quotient algebra $T(W)/I(W, q)$.

Essentially, we take $T(W)$ and "divide out" the ideal $I(W)$. This filters out redundant information. More specifically, when we quotient by this ideal, we are asking that all expressions in $I(W)$ equal zero, which enforces the constraint $w \otimes w = -\langle w, w \rangle \mathbf{1}$ for all vectors $w$. From this, we can derive the anticommutation relations we will use for distinct vectors: $uv + vu = -2\langle u, v \rangle \mathbf{1}$ (for basis elements, this gives $e_i \cdot e_j + e_j \cdot e_i = -2\langle e_i, e_j \rangle$). More details are available in [35, 36].

**One choice of Clifford Algebra representation.** We have described Clifford algebra as an abstract object, but we need a way to work with it for graph encoding. A representation of a $K$-algebra is a homomorphism that maps algebra elements to linear operators. Such a representation allows us to work with Clifford algebra elements as concrete matrices rather than abstract objects. For our practical setup, we will use the formal blueprint this algebra gives, which connects its generators to creation and annihilation operators.

**Practical considerations in Clifford Algebra operations.** Following Clifford algebra axioms exactly allows us to build higher-order elements (like edges, hyperedges) while preserving graph structure. However, a full implementation faces practical issues: an $n$-dimensional space requires a $2^n$-dimensional Clifford algebra, and this is impractical. Therefore, we make design choices that balance rigor with computational feasibility.

## 2.2 From Graphs to Clifford Algebra to Fock Spaces

**Laplacian and the Dirac operator.** For graph $G$, consider the Laplacian $\Delta = II^T \in \mathbb{R}^{|V| \times |V|}$, where $I$ is the incidence matrix [37]. But $\Delta$ represents only one component of the Hodge Laplacian $\mathcal{L}$, which acts on the full exterior algebra of the graph. This captures relations between all grades: scalars (grade-0), nodes (represented as grade-1 vectors), edges (as grade-2 bivectors), and so on. Here, the Dirac operator $D$ serves as a "square root" of the Hodge Laplacian [38], satisfying $D^2 = \mathcal{L}$ [37]. Because the Dirac operator and the graph Laplacian are connected through this identity, we can use the algebraic machinery associated with Dirac operators, i.e., Clifford algebra, for our graph encoding problem.

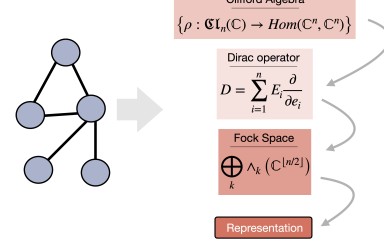

Figure 3: From graph to Fock space representations.

**Connection to Clifford algebra.** The Dirac operator arises naturally from Clifford algebra [39]. Recall that the Clifford algebra is *generated* by $n = |V|$ abstract basis vectors, which we denote as $\{e_1, \ldots, e_n\}$. These are the grade-1 elements of the algebra. The full structure (dimension $2^n$), is then built up from these generators through repeated application of the Clifford product, yielding elements of all grades up to the top grade-$n$ element. One feature of this algebra is the anticommutation relations that these generators satisfy: $e_i e_j + e_j e_i = -2\langle e_i, e_j \rangle \mathbf{1}$ for basis elements. This algebraic structure guarantees the $D^2 = \mathcal{L}$ property holds [40, 36].

**Spinors and Fock space.** We now need to identify the space on which this algebra acts. For the complex Clifford algebra, there exists a special irreducible representation on a complex vector space

$\mathbb{S}$ of dimension $2^{\lfloor |V|/2 \rfloor}$, called the Spinor space [40, 36, 41]. This Spinor space is where the abstract Clifford algebra elements, the generators $e_k$ and their products become concrete operators. However, for graph encoding purposes, we need to handle grades. The Fock space $\mathbb{F} = \bigoplus_{k=0}^{\lfloor |V|/2 \rfloor} \wedge^k (\mathbb{C}^{\lfloor |V|/2 \rfloor})$ provides what we want: a graded structure that holds objects of different types concurrently: scalars (grade-0), nodes (as grade-1 vectors), edges (as grade-2 bivectors), and so on. The key point is that for the Spinor space $\mathbb{S}$ and the Fock space $\mathbb{F}$, there exists an isomorphism $\mathbb{S} \cong \mathbb{F}$ between them [40, 36]. This identification is a link: the Clifford algebra's action is defined on $\mathbb{S}$, and the isomorphism allows us to translate this action onto $\mathbb{F}$, which is the graded "container" of our graph elements.

**Why is this useful?** We can now ask: what do the generators $e_k$ of our Clifford algebra look like when they act on this graded container? When the abstract Clifford generators $e_k$ act on the Fock space, they decompose into two operations: a creation operator $\tau_k^*$ and an annihilation operator $\tau_k$. That is, $e_k \mapsto \tau_k + \tau_k^*$ [40]. The creation operator $\tau_k^*$ increases the grade of an element (e.g., acting on grade-0 to create a grade-1 vector, or acting on a grade-1 vector to form a grade-2 bivector), while the annihilation operator $\tau_k$ decreases the grade. Here, the Dirac operator itself is combining all these creation and annihilation operations: $D = \sum_k (\tau_k + \tau_k^*)$ [36]. The main takeaway for our graph encoding is that the algebraic structure underlying our graph is one of creation and annihilation. Nodes (grade-1) combine to form edges (grade-2), which in turn can combine to form higher-order structures. Unfortunately, this is intractable due to $2^{\lfloor |V|/2 \rfloor}$ dimensions.

## 2.3 Translating Theory to Practice: Instantiating a Graph Representation

The Fock space formulation provides ideas for representing multi-particle systems, viewing particles as nodes in a graph. As noted above, implementing the full structure, in high dimensions, is infeasible. Vector Symbolic Architectures (VSA), as explored in recent works [42, 43], offer a practical approximation of Fock spaces with compute efficiency. In VSA, the binding operation (circular convolution) approximates the creation/annihilation operators, while the superposition operation (vector addition) resembles the direct sum in Fock spaces. Although the VSA $\leftrightarrow$ quantum mechanics connection is not new [44], in this context, it provides specific ideas for efficiency.

**Representing nodes, sums, and products.** In our implementation, we assign a high-dimensional vector to each concept (node, edge, and so on). These vectors are analogous to the basis elements above. While ideally, these vectors would be orthogonal, similar to the properties of basis elements in a Fock space, we simply approximate this by sampling from a normal distribution $\mathcal{N}(\mathbf{0}, 1/d)$. This leads to nearly orthogonal vectors, with the maximum absolute cosine similarity between any two vectors typically below $0.1$ [45].

To emulate operations in Fock space, we use dimensionality-preserving operations instead of tensor products, avoiding exponential growth in dimensionality [44]. This ensures all embeddings maintain the same dimensionality. We define sum ($\oplus$) as element-wise addition and product ($\otimes$) as circular convolution, analogous to Fock space's creation/annihilation operators. Circular convolution is done via element-wise multiplication in the Fourier domain followed by an inverse Fourier transform. As $d$ increases, these operations asymptotically satisfy Fock space's algebraic properties, with complexity $\mathcal{O}(d \log d)$. This framework also supports inverse vectors, where $a \otimes b = \mathbf{1}$. Other properties like commutation relations, superposition, and self-commutation are mostly satisfied. Note that our experiments are not tied to this specific implementation (improved choices can be dropped in).

**Dealing with infinitely many concepts.** In some datasets, vertices include text descriptions, making random vector initialization unsuitable. To address this, we use text encoders like CLIP [24], BERT [46], and RoBERTa [47], which map text to vectors that preserve information and place similar sentences in close proximity. This approach allows us to: **(1)** generate infinitely many vectors, and **(2)** ensure similar vectors represent similar concepts. When dimensionality allows, we retain default sampling and explicitly note the use of text encoders in our experiments.

**Other works using Vector Symbolic Architectures.** Vector Symbolic Architecture (VSA), rooted in symbolic AI, leverages high-dimensional representations alongside logical rules for combining symbols/vectors [48, 49]. Many studies mechanistically derive ways to construct symbols and implement merge operations. The use of Fock space for symbolic manipulation has been explored, with applications in trajectory analysis [44]. Additionally, VSAs have been employed for computational efficiency in self-attention calculations as seen in HRRFormers [50, 42], while a preliminary investigation of the application of VSAs to graphs can be found in [51].

# 3 Fock Graph Encoder (FoGE)

Based on the concepts from §2, we use a parameter-free scheme (denoted FoGE) to obtain rich graph embeddings. Our approach is general and can handle a large spectrum of different graph types, and its extension to novel graph-types is straightforward. Diverse graph types such as hypergraphs, attributed graphs, as well as

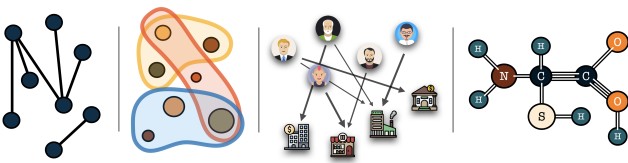

Figure 4: Graphs, Hypergraphs, Attributed graphs, Proteins. All these types can be efficiently encoded using FoGE.

proteins (Figure 4) can all be modeled easily providing an alternative or a good initialization for more intensive trainable models. This approach translates the concepts of Fock spaces into a practical/efficient method for graph representation, where graph features obtained by the encoding are analogous to multi-particle states in a Fock space.

For a graph $G = (V, E)$ we have a set of vectors $[\mathbf{p}_i]_{i=1}^{n=|V|}$, using $i$ to index the nodes. We also use an extra vector $\mathbf{s}$ for the graph's size, a practical design choice we will explain shortly. Then, with these $n + 1$ vectors, we obtain a lossless Fock-space based representation $\mathbf{g}$ as:

$$\mathbf{g} = (\mathbf{s} \otimes \mathbf{p}_n) \oplus \bigoplus_{(i,j) \in E} (\mathbf{p}_i \otimes \mathbf{p}_j) \tag{1}$$

Our formulation follows from §2. Each edge's endpoints are fused together using $\otimes$ and then we aggregate all edges together using $\oplus$. Finally, the graph's size is also added using the special vector $\mathbf{s}$.

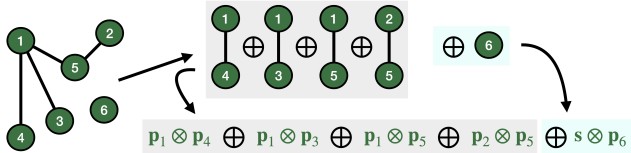

**Lossless representation.** The above representation is lossless. Assuming we use Equation (1) to get a graph's embedding $\mathbf{g}$. Then, simply by evaluating the expression $\mathbf{p}_j^T(\mathbf{p}_i^{-1} \otimes \mathbf{g})$, we can determine whether the edge $(i, j)$ exists in the edge set of that particular graph. In this way, we can recover, one by one, all edges of the graph and correctly reconstruct it, if desired. It is instructive to check the importance of $\mathbf{s}$. By evaluating the expression $\mathbf{p}_i^T(\mathbf{s}^{-1} \otimes \mathbf{g})$, $\forall i$, we can first obtain the size of the graph. This can inform the edge retrieval above because an expression of the form $\mathbf{p}_{n+x}^T(\mathbf{p}_i^{-1} \otimes \mathbf{g})$ could, in practice, produce a number close to 1, although there is no such edge. By first obtaining the size of the graph, we have a "safeguard" against such phantom edges beyond the real vertex-set.

**Vertex attributes.** Consider a graph $G = (V, E, Attr)$, where the set $Attr$ (with $|Attr| = |V|$) consists of attributes, one for each vertex. There is no restriction on the type of attributes: it can denote numerical values or text or any other concept. Let $\mathbf{a}_i$ be the vector associated with the attribute of vertex $i \in V$ (using an appropriate text-encoder if needed). Then, we can augment Equation (1) to absorb the extra information in the following way:

$$\mathbf{g} = (\mathbf{s} \otimes \mathbf{p}_n) \oplus \bigoplus_{(i,j) \in E} (\mathbf{p}_i \otimes \mathbf{p}_j) \oplus \bigoplus_{i \in V} (\mathbf{p}_i \otimes \mathbf{a}_i) \tag{2}$$

The graph is again, fully reconstructable. We have also encoded each vertex's attribute (which can be recovered by the expression $\mathbf{a}_j^T(\mathbf{p}_i^{-1} \otimes \mathbf{g})$). We should think of proteins as a graph with vertex attributes where each vertex is a specific amino acid (possibly with 3-D coordinates).

**Hypergraphs (Theory versus Practice).** Hypergraphs are generalizations of graphs: each edge is connected to an arbitrary number of vertices, instead of just 2 (Figure 4). In theory, we can easily augment Equation (1) so that we can handle hypergraphs as follows:

$$\mathbf{g} = (\mathbf{s} \otimes \mathbf{p}_n) \oplus \bigoplus_{(k_1, \cdots k_m) \in E} \bigotimes_{i=1}^{m} \mathbf{p}_{k_i} \tag{3}$$

In practice, aggregating many multiple vectors together may be unstable. This is true for our particular design choices for calculations (e.g., circular convolution), so we use an alternative approach. We

can start by observing that each edge can be interpreted as a unique cluster of vertices, so we simply assign a unique vector $\mathbf{e}_i$, $i \in \left[|E|\right]$ to each edge in the hypergraph. This modification allows us to encode the hypergraph similar to how a graph is encoded as a dictionary, in the following way:

$$\mathbf{g} = \left(\mathbf{s} \otimes \mathbf{p}_n\right) \oplus \left(\bigoplus_{i=1}^{|E|} \left(\mathbf{e}_i \otimes \bigoplus_{j \in E_i} \mathbf{p}_j\right)\right) \tag{4}$$

### 3.1 Fock Space-based grounding of LLMs (FoGE-LLM)

Recent works showed that (a) textualizing a graph and pre-appending it to a question results in better-than-random responses from the LLM (although far from perfect), and (b) using a specialized graph encoder such as a GNN or a graph transformer and training along with a frozen LLM results in a big improvement in performance, resulting essentially in LLMs that can understand, to some extent, graphical structures. One takeaway is that we can bypass the most tedious stage of designing application-specific graph encoders. Instead, we can use a **parameter-free** method for a wide range of graph types, as we described above. Thus, the *only* trainable parts of the pipeline are simple linear adapters

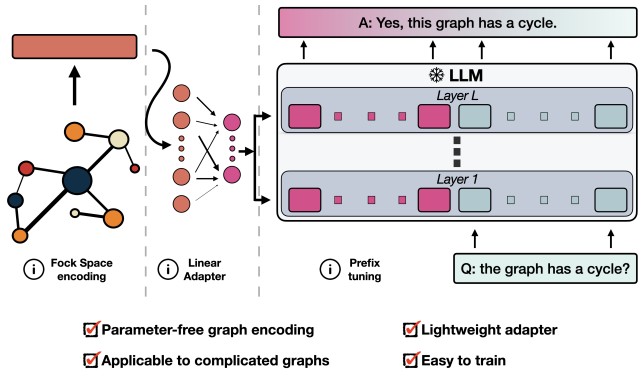

Figure 5: FoGE-LLM overview. Using a parameter-free graph encoder we get graph embeddings for a range of different graphs. Then, we use linear adapters with a frozen LLM for *prefix tuning*.

that convert the raw graph encodings to a format "understandable" by an LLM. Our FoGE-LLM is shown in Figure 5. After getting the graph encodings, we train one/more linear adapters and append the transformed encodings to the question's embeddings fed to the LLM.

**Summary and Takeaway.** We highlight some key advantages. *First*, our graph encoding is parameter-free and efficient. The complexity of aggregation is $\mathcal{O}(d \log d)$ ($d$ is vectors' dimension) and the number of aggregation operations is linear (in graph size). *Second*, our encoder is not restricted to specific graph types: it works easily for simple graphs, for proteins and for hypergraphs just via small modifications. In contrast, GraphToken [16] uses a specific GNN whose output size is dependent on the underlying task whereas GraphLLM [17] uses a transformer model together with a GNN (also specific to the underlying task). These properties simplify our training and eliminates any tunable components. *Third*, our open-source code offers a scalable way to train FoGE-LLM even on consumer GPUs, by using FSDP [52]. For reference, GraphToken [16] is trained on TPUs (code unavailable) whereas GraphLLM [17] has a large memory/compute footprint (trained on A100 80GB).

## 4 Experimental results

We examine our Fock-space based encoding in two separate settings: (a) as a stand-alone input of a simple model, and (b) as an extra prefix in a frozen LLM (FoGE-LLM), for graph prompting. Our initial experiments (§4.1) show that simple models can process FoGE embeddings quite well for traditional tasks. In §4.2, we present that FoGE can be successfully combined with an LLM, leading to two advantages over stand-alone models: (1) language interface for flexible graph queries without pre-defining task types, and (2) easier integration with mature software ecosystems built around LLMs that reduce deployment overhead.

**Datasets and Models.** We performed experiments on multiple graph reasoning datasets: from simple graph-understanding tasks to hypergraphs and proteins and aim to cover different aspects of graph understanding/reasoning. Specifically, we consider the 7 following datasets/dataset collections: **(i) GraphQA** [15] **(ii) GraphReasoning** [17] **(iii) HyperGraphQA (iv) PPI** [53] **(v) OBNB** [54] **(vi) mol-HIV** [55] **(vii) SabDab** [56]. More details about the datasets can be found in the appendix. Exploring diverse graph reasoning datasets helps evaluate our model's performance and

Table 1: Using a small neural network with a single layer on the obtained graph representations allows us to perform near-perfectly on tasks such as *number of nodes* and *number of edges*, for both synthetic and real data.

| | GraphQA | | | HyperGraphQA | | Jaffe | |
|---|---|---|---|---|---|---|---|
| | num nodes | num edges | has cycle | num nodes | num edges | num acids | num links |
| MSE/Acc | 0.67 | 0.03 | 98.7% | 1.12 | 0.63 | 2.95 | 11.9 |
| Model size | 32K | 8K | 16K | 32K | 4K | 32K | 16K |

generalization across various graph structures and domains, from traditional graph-based QA to hypergraph understanding and biological network analysis. By including datasets like PPI and BioGRID, we seek to check the practical relevance of our result, with potential applications in biology, network analysis, and more. We use the Llama2 (7B) model [57] as the frozen LLM, and we use only extra linear adapters for the graph embeddings obtained using our formulation. We adjust vector dimensionality from 512 to 2048 and use just a *single adapter* for the entire model or *one adapter per layer* in FoGE-LLM.

## 4.1 Proof of Principle Evaluations for Graph Understanding

**Setup and Results.** While our key goal is graph-prompting, we first perform multiple preliminary checks of the effectiveness of our graph encoding. We conduct three different types of experiments.

*First*, we evaluate whether our graph embeddings are informative (i.e., they preserve the graph's structure), by using a small, 1-hidden-layer FFN for basic graph-understanding tasks. The results in Table 1 show that our representations are rich and informative. More specifically, we assess the quality of the embeddings by first encoding graphical structures from 3 different classes of graphs (graphs, hypergraphs, and proteins) and then training a small model (one hidden layer) to predict graph attributes like *number of nodes* and *edges*. The results indicate that our representations are rich and informative and only few parameters suffice to achieve almost-perfect performance on such tasks.

Table 2: Results on mol-HIV [55, 58]. The full details can be found on https://ogb.stanford.edu/docs/leader_graphprop

| | ROC-AUC |
|---|---|
| HyperFusion | 0.8475 ± 0.0003 |
| PAS + Fingerprint | 0.8420 ± 0.0015 |
| HIG | 0.8403 ± 0.0021 |
| DeepAUC | 0.8352 ± 0.0054 |
| **FoGE** + Fingerprint | 0.8305 ± 0.0068 |
| GMAN + Fingerprint | 0.8244 ± 0.0033 |
| RF + Fingerprint | 0.8208 ± 0.0037 |
| **FoGE** | 0.7614 ± 0.0051 |
| GCN | 0.7606 ± 0.0097 |
| GIN | 0.7558 ± 0.0140 |

Additionally, for more involved tasks, we consider the Open Graph Benchmark (OGB) [58] (more specifically, the mol-HIV[55] dataset) and we show that our encoding is better than multiple, heavy, specialized, and trainable methods while being training-free and unsupervised! In Table 2, we show the results. Interestingly, because our method allows a seamless integration of additional graph information (like a molecule's footprint), we find that this can help us achieve even better results and, in some cases, be competitive with submissions at the top of the leaderboard. To obtain the final results, we used AutoGluon [61] on our unsupervised embeddings, with a time limit of 10 minutes, similarly to the strategy of many of the other baselines.

Table 3: Results on two real protein datasets from OBNB. Our method is the strongest unsupervised scheme to obtain node embeddings, especially for DisGeNet. Its performance is comparable to trainable, graph-specific models. More details on all baselines are in [54]. The reported metric is the APOP (Average test Precision Over Prior).

| | BioGRID | | HumanNet | |
|---|---|---|---|---|
| Model | DisGeNet | GOBP | DisGeNet | GOBP |
| LabelProp | 0.931 | 1.885 | 3.059 | 3.806 |
| Adj + LR | 0.743 | 2.528 | 3.053 | 3.964 |
| Node2Vec + LR | 0.836 | 2.571 | 2.433 | **4.036** |
| LapEigMap + LR | 0.864 | 2.149 | 2.301 | 3.778 |
| **FoGE** | **1.062** | 2.433 | **3.254** | 3.916 |
| GCN [59] | 1.012 | **2.572** | 3.116 | 3.812 |
| GAT [60] | **1.063** | 2.562 | 3.065 | 3.963 |

*Second*, we examine whether our graph encodings preserve important biological markers of the data. To test this, we use a small dataset of about 900 proteins (SabDab [56]) which are accompanied by affinity data that corresponds to each protein's clade. Briefly, clades are protein superfamilies, based on common ancestry (more information can be found in the appendix). In principle, proteins from the same clade are *more similar* than across clades, so we examine whether this is also preserved

in our obtained embeddings. Although the dataset has only few samples and some of the clades are scarcely populated, we can observe that there is a clear separation between the most populated clades in the embeddings space.

*Third*, we examine if the same encoding practice can generate rich node-level encodings, by encoding for each node, the subgraph that is generated by itself and its neighbors. We examine performance on **nineteen** real protein datasets (OBNB [54] and PPI [53]). The detailed results on the complete OBN Benchnark can be found on the appendix (4 of them are presented in Table 3), while Table 4 demonstrates FoGE's performance on PPI. We see that our approach is, in all datasets, among the best unsupervised approaches, and is also competitive (if not better) than specialized supervised approaches that leverage trainable, graph-specific models such as GCN [59] and GAT [60]. Specifically, we achieve state-of-the-art performance in PPI. We also achieve the best results (among both unsupervised and supervised) in seven out of the eighteen datasets of OBNB.

These results provide encouraging evidence that (a) our approach gives "rich" graph embeddings for a range of different graph types and styles, and (b) our graph embeddings can be used as an extra, grounding input to a powerful LLM without the need to design/train a specialized model, e.g., GNN [67, 68] or a Graph Transformer [69].

Table 4: Micro F1-score on PPI. Our approach is better than the best unsupervised approaches and better/comparable to the supervised approaches.

| | Model | F1 |
|---|---|---|
| | Random | 39.2 |
| | Node2Vec [62] | 40.9 |
| | Raw features [62] | 42.2 |
| Unsupervised | GraphSAGE-min [53] | 46.5 |
| | GraphSAGE-max [53] | 50.2 |
| | DGI [63] | 63.8 |
| | GRACE [64] | 66.2 |
| | **FoGE** | **99.2** |
| Supervised | GraphSAGE-min [53] | 50.0 |
| | GraphSAGE-max [53] | 61.2 |
| | LGCN [65] | 77.2 |
| | GAT [60] | 97.3 |
| | GCNII [66] | **99.5** |

## 4.2 Grounding LLMs with Graph prompting

We next investigate whether such an encoder can be successfully integrated with an LLM. Can an LLM understand graph structure? Our experimental evidence suggests that it can: the model learns to reason about graph properties and, in many cases, performs better than approaches relying on heavily specialized graph encoders. While this line of research is still in its early stages and requires further exploration, it closely parallels recent multimodal advances. Just as LLaVA [70, 71, 72] acquires visual understanding and TimeLLM [73] develops temporal reasoning once their respective modalities are properly embedded, our findings indicate that LLMs can develop some graph reasoning capabilities when provided with rich structural representations such as those produced by FoGE.

**Graph Understanding.** In our first experiment, we examine whether an LLM can answer questions about a graph's structure, such as the number of nodes, the presence of cycles, and so on. We use GraphToken and conduct a suite of six different experiments. Although our method's encodings are *not* specific to each underlying task, it performs competitively with specialized models, as shown in Table 5. Even when GraphToken uses different embeddings for each node (*node degree*) or edge (*edge existence*), our model still achieves comparable results using a single embedding for the entire graph.

Table 5: GraphToken vs FoGE-LLM on GraphQA. Column *1* stands for a single embedding for the entire graph; $\mathcal{O}(n)$ stands for a single embedding per node. In all 6 tasks, although we use a parameter-free, predetermined graph encoding, we see a performance similar/better relative to a trainable graph encoder with a larger LLM.

| | ICL | GraphToken | | **FoGE-LLM** |
|---|---|---|---|---|
| Tokens | $\mathcal{O}(n^2)$ | 1 | $\mathcal{O}(n)$ | 1 |
| num of nodes | 26.9% | **99.6%** | - | 97.2% |
| num of edges | 12.8% | 42.6% | - | **45.1%** |
| cycle existence | 83.2% | 95.6% | - | **97.9%** |
| num of triangles | 16.2% | 34.8% | - | **37.7%** |
| edge existence | 54.4% | - | 73.8% | **74.3%** |

**Advanced Graph Reasoning.** Going beyond "simple" graph understanding tasks, we also examine our performance on more complicated graph-reasoning tasks, using the GraphReasoning dataset [17]. GraphToken is not applicable here since each node is accompanied by a textual description which cannot be handled by that model. So, our main baseline is GraphLLM, which uses a transformer combined with a GNN to merge the graphical/textual information into one or more embedding vectors. Similar to GraphToken [16], GraphLLM [17] also utilizes a different approach for each task, using multiple graph embeddings for each task. In contrast, we achieve comparable performance using a *single graph embedding*, showcasing the versatility/richness of the graph embeddings (Table 6).

Further, we see that using a pretrained text encoder such as RoBERTa [47] to generate the vectors is reasonable, and results in a similar performance. This is a strong improvement over traditional symbolic methods, by allowing a large set of "symbols"/vectors. Dealing with proteins is similar to advanced graph reasoning, since both datasets are graphs with additional node information.

Furthermore, in Table 7, we show the accuracy of FoGE-LLM for three protein-related tasks on Jaffe. Although the size of the protein graphs is more than $10\times$ larger compared to the ones in GraphQA and GraphReasoning, our model is able, up to some extent, to understand the provided protein, as a whole (*number of amino acids* and *number of links*) as well as at an individual-node level for the task *type of amino acid* (where we prompt the model to determine the type of a specific vertex in the protein).

**Hypergraphs.** Existing works focus on specific forms of graphs and rarely applicable (or easily modifiable) to different graph types. One common family of graphs in applications is hypergraphs.

Table 6: GraphLLM vs FoGE-LLM. Although we are using the same, predetermined graph embedding for each task, we enjoy a performance similar to GraphLLM which leverages 5 graph embeddings, specific to the task at hand. The *vectors* stands for the two approaches we follow in generating them: (a) randomly generated (almost) orthogonal vectors (ignoring the node's text description), and (b) using RoBERTa [47] and utilizing all vertices' information.

|  | GraphLLM | FoGE-LLM | |
|---|---|---|---|
| model size | 100M | 25M | |
| question specific output | Yes | No | |
| graph embeddings | 5 | 1 | |
| vectors | - | random | RoBERTa |
| substructure count | 99.9% | 97.3% | 95.6% |
| max triplet sum | 95.7% | 94.6% | 94.7% |
| shortest path | 97.2% | 95.7% | 95.8% |
| bipartite match | 99.8% | 98.1% | 97.3% |

Here each edge is a subset of the nodes, of arbitrary size (Figure 4). Our formulation can handle such a generalization of the typical graphs with only minor modifications to the encoding formulation (Equation (4)). Here, we show that our setup can indeed answer questions about such complicated structures, using our encodings as an extra prefix (graph prompting). Using the HyperGraphQA dataset, we assess the performance of FoGE-LLM on four common tasks. Since GraphToken as well as GraphLLM cannot handle such data, we compare our model's performance against two of the most common prompt-engineering methods: 1. zero-shot, where the model is given the graph in text form along with the corresponding question, and 2. few-shot, where the model is given pairs of textualized graphs with the corresponding question/answer pair and it is asked to produce the answer to a new combination of graph/question. The results are presented in Table 7. Interestingly, even though hypergraphs have a much more complicated structure than "simple" graphs, our model achieves a performance very close to basic graph understanding (Table 5), or even better at some tasks.

### 4.2.1 FoGE-LLM runtime

Besides the raw performance gains as presented above, FoGE-LLM offers a very low inference time, due to two reasons. First, we always "reserve" only a single token for the given graph. In contrast, zero/few-shot approaches that textualize the graph require a large number of tokens, and grows larger as the graph grows. This leads to an increase in inference time, due to the number of input tokens. Second, FoGE-LLM uses one or more linear adapters, no specialized architectures like in [17, 16] are needed. Based on our experiments, this impacts inference time,

Table 7: FoGE-LLM performance against ICL techniques for hypergraphs and proteins.

|  |  | Zero-Shot | Few-Shot | **FoGE-LLM** |
|---|---|---|---|---|
| HyperQA | num of nodes | 04.5% | 16.8% | **85.0**% |
|  | num of edges | 03.9% | 27.0% | **95.4**% |
|  | node degree | 02.1% | 10.1% | **53.9**% |
|  | edge existence | 65.9% | 79.4% | **87.9**% |
| Jaffe | num of amino-acids | 03.9% | 17.1% | **99.3**% |
|  | num of links | 03.8% | 06.1% | **13.2**% |
|  | amino-acid type | 01.4% | 12.3% | **37.7**% |

and gives FoGE-LLM strong efficiency benefits. In Table 8 we show the average inference time required for each approach.

## 5   Related Work

**Geometric Algebra in Machine Learning.** There is growing interest in application of geometric algebra in machine learning, particularly for developing models that maintain geometric properties.

While these ideas have been leveraged in the context of equivariance/symmetry transformations in deep learning [74, 75, 76, 77, 78], the concept is finding interesting uses in recent works. For example, [79] recently proposed Clifford Neural Layers to model dynamical systems in fields like fluid dynamics and [29] described Geometric Clifford Algebra Networks (GCANs), specifically designed to respect symmetry group transformations. Beyond classical machine learning, geometric algebra finds more direct applications in quantum computing as well: [80] leveraged the isomorphism between Pauli matrices and Clifford Algebra to represent multidimensional data, to define specialized transforms for machine learning tasks.

**Graphs & LLMs.** The body of work describing ways to infuse extra, graphical information into a frozen LLM is sizable and growing. As discussed earlier, initial approaches focused on converting the underlying graph into natural language form, such as "node 1 is connected to node 3, node 5 is connected to node 4, ..." [13, 14, 15]. These works while far from perfect showed viability: that a frozen LLM has the capability to reason about the given graph and answer graph-related questions, such as "is there a cycle in the graph?". Practical difficulties involving the format of graph serialization is an important factor in the performance and the results tend to be only moderately better than random. The perspective taken in [16, 17] was fresh and led to an alternative approach: infusing the graph

Table 8: Average inference time for each approach. FoGE-LLM is significantly lower than zero/few shot approaches since the number of input tokens does not grow with the graph size, while it enjoys a 40% improvement over GraphLLM dues to its simpler encoder/adapter.

| Model | Inference time (s) $\downarrow$ |
|---|---|
| zero-shot | 0.175 ($\pm$0.05) |
| few-shot | 0.541 ($\pm$0.10) |
| GraphLLM [17] | 0.052 ($\pm$0.01) |
| **FoGE-LLM** | 0.031 ($\pm$0.01) |

information directly at the embedding level, by encoding the graph using a model such as a Graph Neural Network (GNN) [67, 68, 16] or a Graph Transformer [69, 17]. These works significantly improved the state of the art, showing that carefully crafted graph embeddings are key to a successful grounding of an LLM.

# 6 Conclusions

We have described a novel scheme to encode a graph into a vector form for direct downstream use or to augment prompts fed to LLMs. Our approach, motivated by Fock space operations, offers numerous advantages in practice demonstrated via experiments. We can obtain encodings of arbitrary graphs quickly, with no trainable parameters, that nicely captures the important information content in the underlying graph. To utilize these encodings, we introduced FoGE-LLM – a way to fuse the graph information for graph-prompting with a pre-trained, frozen LLM, allowing it to "understand" and reason about graphs. Given the growing interest in grounding LLM responses based on additional domain-specific priors, we believe that this is an interesting direction. Our model, accompanied with a simple-to-train open-source codebase, performs favorably relative to highly specialized models. It is also quite flexible and can handle classes of graphs where other alternatives fall short or need adjustments.

**Impact & Limitations.** A key strength of our method is its parameter-free approach for generating rich graph embeddings. Such an approach can be a great fit in less computationally rich environments or in cases where the dataset's size is not big enough for the trainable approaches, without, as we demonstrated extensively, lacking in performance in data-rich situations. Given the scarcity of the data in many real-life graph-related problems (like the protein-based questions we answered here), our approach can benefit multiple aspects of research. However, the unsupervised nature of FoGE also limits the ability to fine-tune performance if the embeddings are insufficient for specific applications. So, we believe that building representation learners on top of these embeddings, as in FoGE-LLM, is a good strategy. Additionally, when dealing with infinitely large vector sets, random generation is impractical. While RoBERTa works well in our experiments, integration with other models may involve some trial-and-error to identify sensible configurations.

**Acknowledgments** We would like to thank Tom Reps and Anthony Gitter for discussions and feedback. Additional experiments on the applicability of FoGE in other types of datasets were performed during a summer internship of S.P. Chytas at LLNL, and will be disseminated in a separate paper after approvals.

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

# A  Dataset details

In our experiments, we used the following datasets:

1. **GraphQA** [15]: It includes 6 different graph-understanding tasks (*number of nodes*, *number of edges*, *cycle existence*, *number of triangles*, *node degree*, and *edge existence*) on 7 different graph structures (Erdos-Renyi [81], Scale-Free, Barabasi-Albert [82], Stochastic Block Model, Star, Path and Complete).

2. **GraphReasoning** [17]: Recently introduced in [17] to better assess the model's graph understanding ability, it consists of 4 more advanced graph-understanding tasks (*substructure count*, *maximum triplet sum*, *shortest path*, and *bipartite graph matching*). Each graph node is accompanied by extra information in the form of a text description, making this dataset a suitable testbed for our RoBERTa-based vector encoding.

3. **HyperGraphQA**: We extend GraphQA to Hypergraphs. Specifically, we consider 4 different graph-understanding tasks (*number of nodes*, *number of edges*, *node degree*, and *edge existence*) on 2 different hypergraph structures (Erdos-Renyi [81], and Chung-Lu [83]). The training dataset consists of only 2000 instances, making it hard for large models to avoid overfitting.

4. **Jaffe** [84]: Jaffe is a recent dataset consisting of approximately 1.6 million natively paired human antibody sequences from healthy donors. To our knowledge, this represents by far the largest publicly available dataset of its kind.

5. **PPI** [53]: PPI consists of 24 proteins collected from human tissue, with each node associated with 121 binary labels. Compiled from experimental techniques like yeast two-hybrid screening and mass spectrometry, as well as computational predictions, such a dataset provides critical insights into the functional organization of the proteome. By understanding how proteins interact, scientists can uncover the molecular underpinnings of cellular processes and develop targeted therapeutic strategies.

6. **OBNB** [54]: OBNB (Open Biomedical Network Benchmark) is a collection of 15 datasets (including well-known datasets such as BioGRID [85] and HumanNet [86]). Each dataset's sample consists of a gene accompanied by 3 vectors (named *DISEASES*, *DisGeNET*, *GOBP*) of node-level binary labels.

7. **SabDab** [56]: SabDab (Structural Antibody Database) is a collection of 919 publicly available, annotated antibody structures (proteins). Each structure is accompanied by multiple annotations, such as the heavy and light chain pairing.

8. **mol-HIV** [55]: The ogbg-molHIV dataset consists of molecular graphs (atoms as nodes, chemical bonds as edges) labeled for the binary classification task of predicting whether a molecule inhibits HIV replication or not. Each molecule is represented with 9-dimensional atom features (e.g. atomic number, chirality, ring membership, formal charge), and the dataset is evaluated using scaffold splits with ROC-AUC as the metric.

# B  FoGE-LLM

## B.1  Training details

We train the LLM-based construction with a batch size of 16 and a learning rate of $1e$-3. The model required less than 10 epochs to convergence, in contrast to other works that require more training time due to the ellaborate graph encoders (e.g., [17]). Our implementation is based on Pytorch Lightning [87], which allows us to split and train the model on multiple GPUs using FSDP. This implementation allows the user to train this, or any similar, model to conventional GPUs with less memory while, at the same time, speed up the process by preloading all the obtained lightweight graph embeddings to the GPUs. The *merging* of the graph embedding with the LLM is based on the idea of prefix tuning [18], i.e., pre-append the embedding to the input text embeddings and, in our case, this is happening with the use of a linear adapter. We experimented both with a single linear adapter on the input layer, as well as a linear adapter per layer and the difference was only marginal in the final results.

## B.2 Inference details

Besides the low training time, FoGE-LLM enjoys an extremely low inference time, due to two reasons. First, we always "reserve" only a single token for the provided graph. In contrast, zero/few-shot approaches that textualize the graph require a large number of tokens, prohibitively large as the graph grows. This leads to an explosion of the inference time, due to the transformer's quadratic dependency on the number of input tokens. Second, FoGE-LLM employs one or more linear adapters and does not require any specialized architectures, like existing solutions [17, 16]. This, as we observed in our experiments, impacts the inference time, casting FoGE-LLM one of the fastest graph-augmented Language Models. In Table 9 we present the average inference time required for each approach.

Table 9: Average inference time for each approach on Llama-7B. FoGE-LLM is significantly lower than zero/few shot approaches since the number of input tokens does not grow with the graph size, while it enjoys a $40\%$ improvement over GraphLLM dues to its simpler encoder/adapter.

| Model | Inference time (s) $\downarrow$ |
|---|---|
| zero-shot | $0.175\ (\pm 0.05)$ |
| few-shot | $0.541\ (\pm 0.10)$ |
| GraphLLM [17] | $0.052\ (\pm 0.01)$ |
| **FoGE-LLM** | $0.031\ (\pm 0.01)$ |

## C  ICL prompting for hypergraphs

In Table Table 7 we demonstrate FoGE's superiority over In-Context Learning approaches, like zero-shot and few-shot prompting. Here we explain how we created the textual descriptions of the hypergraphs, that were used in both zero- and few-shot prompting. Following similar works for graph textualization [16, 15], we first assign a number to each node and then, in a new line, we explain which nodes are part of each hyperedge. An example can be seen below.

> G describes a hypergraph among 0, 1, 2, 3, 4, 5, 6, 7, and 8.
> In this hypergraph:
> Hyperedge 1 connects nodes 2, 3, 6.
> Hyperedge 2 connects nodes 1, 4, 5, 7.
> Hyperedge 3 connects nodes 1, 2.
> Hyperedge 4 connects nodes 3, 5, 7, 8.

After the hypergraph textualization, the question follows in the case of zero-shot, while both the question and the answer follow in the case of few-shot.

## D  Lossless representations

One advantage of the obtained embeddings is that fact that the underlying structures are recoverable. This allows us to obtain unbiased vector estimates of complicated structures, such as graphs with multiple edge and node attributes. Here, we show how this property manifests in our specific formulation as well as more generally for pairs of key-item.

### D.1  Capacity

One of the typical ways to examine the performance of such a construction is by assuming a vector $\mathbf{u}$ as being the bundling of multiple binded pairs, as in the following equation

$$\mathbf{u} = \bigoplus_{i=1}^{n} \mathbf{k}_i \otimes \mathbf{v}_i \tag{5}$$

and then examine how accurately we can recover each vector $\mathbf{v}_i$, given the corresponding $\mathbf{k}_i$. In theory, the vector $\mathbf{v}_i$ can be easily recovered using the operation:

$$\tilde{\mathbf{v}}_i = \mathbf{k}_i^{-1} \otimes \mathbf{u} \tag{6}$$

In Fig. 6 we examine the cosine similarity of the obtain vector $\tilde{\mathbf{v}}_i$ with the correct one ($\mathbf{v}_i$) as well as with all the rest ($\{\mathbf{v}_j\}_{j \neq i}$). We observe that the results follow closely the theoretical results above, with a perfect separation of up to 100 pairs, and a small overlap for $200 - 300$ pairs of vectors.

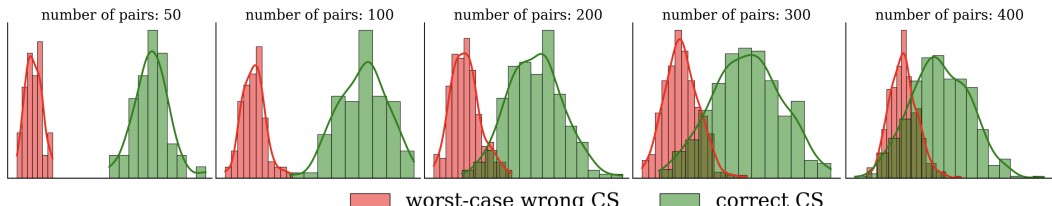

Figure 6: Given a vector $\mathbb{R}^{4096} \ni \mathbf{u} = \oplus_{i=1}^{n} \mathbf{k}_i \otimes \mathbf{v}_i$, how correctly we can recover all pairs of keys-values back, as the number of pairs ($n$) grows. *Worst-case wrong CS* corresponds to the maximum cosine similarity of the recovered value vector with all value vectors but the correct one, and *correct CS* corresponds to the cosine similarity with the correct value vector.

## D.2    Graph reconstruction

In our specific application, we deal with graphs and, as we analyzed, the graph representations we obtain are, in theory, lossless, i.e., we can recover back the original graph from the vector representation using the inverse vectors. Here, we examine whether this claim holds in practice too. In Fig. 7 you can observe the strength of each edge after reconstruction, for 3 different vector dimensionalities. We can observe that, even for a moderately large dimension, there is a clear separation between the true edge set and the rest of the edges.

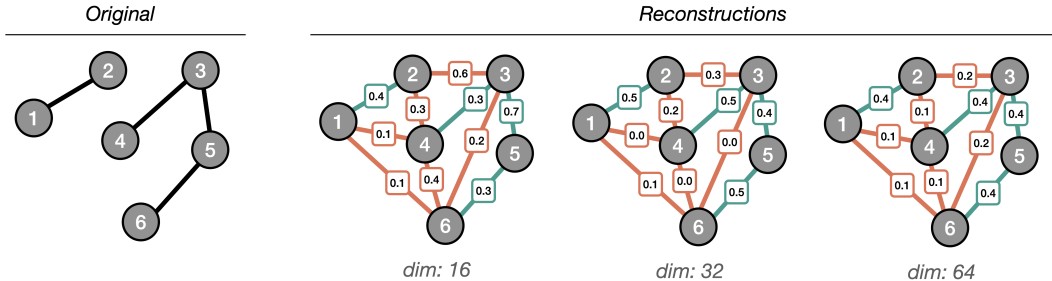

Figure 7: Lossless representations: even for small vector dimension, we can obtain back the true edge set. The numbers show the cosine similarity of the obtained vector with the true edge vector, and it can be used to estimate the true edge set.

## E    FoGE runtime

In Table 10, we show the runtime of FoGE as we increase the number of edges, on a conventional consumer CPU. The graphs are randomly generated instances of Erdos-Renyi graphs. From this experiment, three observations stand out:

1. The linear relationship between the number of edges and the runtime.
2. Fast encoding of graphs with even millions of edges.
3. The ability of FoGE to handle huge graphs without any substantial increase in memory consumption.

Additionally, given the properties of the aggregation we use, the operations can be parallelized on multiple CPU threads, speeding up even further the computation on larger graphs. For example, similarly to our experiments with OBNB, we can calculate in parallel a single embedding per node, and then aggregate them together. This can lead to significant improvement in encoding, exploiting the fact that our aggregation operations are lightweight and can even run on CPUs.

| number of edges | 1 | 500 | 50000 | 1200000 | 5000000 |
|---|---|---|---|---|---|
| runtime (sec) | 0.0006 (±0.00) | 0.041 (±0.01) | 3.769 (±0.35) | 94.041 (±1.04) | 378.018 (±4.55) |

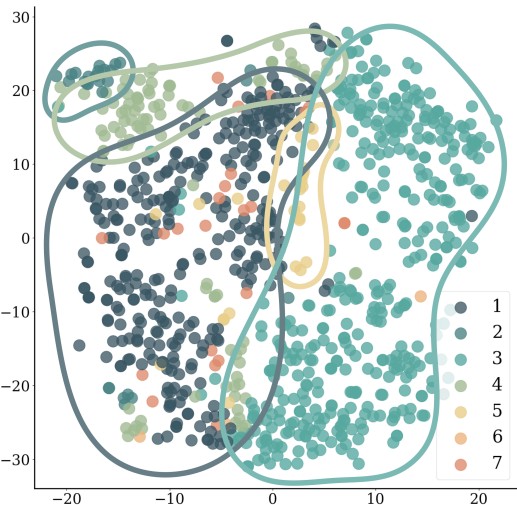

Figure 8: T-SNE plot of the SabDab embeddings. Although the dataset is very small, each one of the populated clades occupies a different region and, interestingly, clades 1 and 7 are very similar, just like in real life. The T-SNE plot was robust to different choices of hyperparameters, with no significant differences beyond simple translations of the space.

## F    Preservation of Clade information on SabDab

Given that the SabDab proteins [56] are annotated with the heavy/light chain pairing, we can extract the clades and visualize their embeddings with respect to that information. As a brief reminder, the clades correspond to superfamilies of proteins that share a common ancestor [88]. To extract the clades we used the V gene heavy chain and chose seven families. It is well known from biology that antibodies that belong to the same clade are *more similar* than antibodies across different clades, so, here, we examine if this real-world, biological property is reflected on our embeddings. Specifically, after obtaining each protein's embedding using FoGE (in an unsupervised fashion without using the clade annotations), we apply a T-SNE transformation on the high-dimensional vectors so that we are able to plot them, with a significant amount of noise, in just two dimensions. Although we reduce the dimensionality significantly, and, even worse, we deal with a extremely small dataset of just 919 proteins (Table 11), in Fig. 8 we can observe that the proteins of each clade cluster together. This is a different, qualitative indicator, which shows that FoGE is able to preserve all the information that is encapsulated in the inputted structures.

Table 11: Distribution of samples across the different clades. In total there are 919 samples, with clades 1, 3, 4 being the most frequent.

| **Clade** | 1 | 2 | 3 | 4 | 5 | 6 | 7 | *Total* |
|---|---|---|---|---|---|---|---|---|
| **Count** | 325 | 28 | 414 | 101 | 25 | 3 | 23 | 919 |

## G    Additional results on OBNB

OBNB (which stands for Open Biomedical Network Benchmark) is a collection of multiple, real-world protein datasets, where each node (or amino-acid) of each protein is accompanied by multiple binary labels. A detailed analysis of the datasets and their labels can be found in [54] and the corresponding repository. In Table 12 we present the results on all 18 reported datasets of OBNB.

Table 12: FoGE vs multiple unsupervised and supervised methods. After obtaining our embeddings, we use a Random Forest to predict the corresponding node's label. The evaluation is based on the APOP metric [54] and we can observe that FoGE is always comparable to the best methods, while in almost half of the cases it is the best one.

| Network | Model | DISEASES | DisGeNET | GOBP |
|---|---|---|---|---|
| BioGRID | LabelProp | 1.210 | 0.931 | 1.858 |
| | LogReg | 1.556 | 1.026 | 2.571 |
| | GCN+BoT | 1.511 | 1.014 | 2.442 |
| | SAGE+BoT | 1.486 | 1.031 | 2.402 |
| | GIN+BoT | 1.410 | 1.007 | 2.386 |
| | GAT+BoT | **1.609** | 1.037 | **2.624** |
| | GatedGCN+BoT | 1.547 | 1.038 | 2.517 |
| | **FoGE** | 1.599 | **1.062** | 2.433 |
| HumanNet | LabelProp | 3.728 | 3.098 | 3.806 |
| | LogReg | 3.812 | 3.158 | **4.053** |
| | GCN+BoT | 3.552 | 3.053 | 3.921 |
| | SAGE+BoT | 3.401 | 3.052 | 3.816 |
| | GIN+BoT | 3.513 | 3.054 | 3.861 |
| | GAT+BoT | 3.761 | 3.100 | 3.809 |
| | GatedGCN+BoT | 3.677 | 3.086 | 3.889 |
| | **FoGE** | **3.853** | **3.254** | 3.916 |
| COMPPIHumanInt | LabelProp | 1.352 | 1.106 | 2.076 |
| | LogReg | 1.644 | 1.240 | **2.806** |
| | GCN+BoT | 1.648 | 1.211 | 2.685 |
| | SAGE+BoT | **1.694** | 1.210 | 2.629 |
| | GIN+BoT | 1.608 | 1.219 | 2.611 |
| | GAT+BoT | 1.665 | 1.230 | 2.755 |
| | GatedGCN+BoT | 1.672 | 1.218 | 2.735 |
| | **FoGE** | 1.660 | **1.241** | 2.586 |
| BioPlex | LabelProp | 0.964 | 0.939 | 1.714 |
| | LogReg | **1.358** | **0.939** | 2.587 |
| | GCN+BoT | 1.324 | 0.911 | 2.553 |
| | SAGE+BoT | 1.246 | 0.865 | 2.513 |
| | GIN+BoT | 1.349 | 0.868 | 2.504 |
| | GAT+BoT | 1.355 | 0.873 | 2.548 |
| | GatedGCN+BoT | 1.301 | 0.859 | 2.590 |
| | **FoGE** | 1.273 | 0.879 | **2.599** |
| HuRI | LabelProp | 0.545 | 0.598 | 1.086 |
| | LogReg | 0.650 | 0.656 | 1.084 |
| | GCN+BoT | 0.634 | 0.693 | 1.129 |
| | SAGE+BoT | 0.593 | 0.679 | 1.190 |
| | GIN+BoT | 0.583 | 0.702 | 1.143 |
| | GAT+BoT | 0.667 | 0.687 | 1.174 |
| | GatedGCN+BoT | 0.596 | 0.695 | **1.195** |
| | **FoGE** | **0.684** | **0.729** | 1.070 |
| OmniPath | LabelProp | 1.358 | 0.897 | 1.593 |
| | LogReg | 1.542 | **1.093** | **2.125** |
| | GCN+BoT | **1.577** | 1.068 | 2.071 |
| | SAGE+BoT | 1.478 | 1.062 | 1.986 |
| | GIN+BoT | 1.452 | 1.073 | 1.993 |
| | GAT+BoT | 1.552 | 1.048 | 2.068 |
| | GatedGCN+BoT | 1.516 | 1.049 | 2.071 |
| | **FoGE** | 1.511 | 1.085 | 2.102 |

FoGE is one of the best-performing methods across all benchmarks, showcasing once more the capabilities of our obtained embeddings.

## H   Impact of vector dimension

One of few the hyperparameters of FoGE is the dimensionality of the vectors (i.e. graph embeddings). Using GraphQA, we perform an ablation study on the impact of the dimension on the final accuracy of the model (Fig. 9). Relative accuracy is calculated as the actual accuracy for each dimensionality,

divided by the best one, for each task respectively, and it allows us to compare different tasks with completely different best performances (Table Table 5).

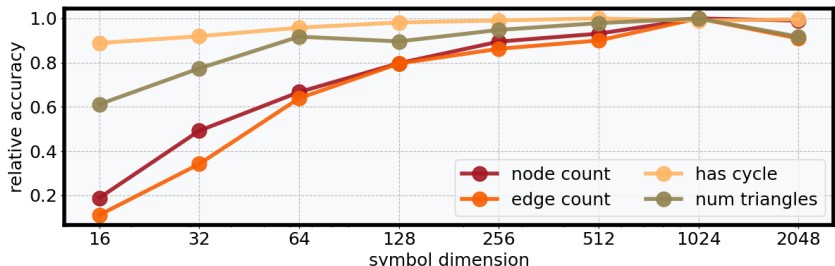

Figure 9: Accuracy versus vectors dimensionality. Although there is a positive trend between the two quantities, the dependency on the dimension is not equally strong or always positive in all tasks.

From this study, a few important remarks surface that we observe to hold true for the other datasets too. First of all, a larger dimensionality does not always "translate" to better results. We observe that for some tasks (*cycle existence*), we achieve the optimal performance with a dimension significantly lower than the maximum we consider (2048), matching essentially GraphToken's performance with less than 20K trainable parameters, while in some cases there is a small drop as we go from 1024 to 2048. Finally, as with most of the tunable hyperparameters in machine learning models, there is no predetermined best strategy for choosing the dimensionality. For instance, when we consider *cycle existence* or *the number of triangles* we can have a highly performing model with a dimensionality of less than 128, while for tasks such as *edge and node count* the performance drops significantly as we reduce the dimensionality.

