# OpenReview forum: "FoGE: Fock Space inspired encoding for graph prompting"
_NeurIPS.cc/2025/Conference — NeurIPS 2025 poster_

### Official Review · Reviewer_nube · 2025-07-03

**Clarity:** 2
**Significance:** 2
**Originality:** 3
**Rating:** 4
**Confidence:** 3

**Summary:**

This paper proposes FoGE, which applies Fock space concepts from quantum mechanics to create a parameter-free graph encoder for LLM prompting. The authors argue this approach can generate rich and informative graph representations that work well across different graph types, from simple networks to complex structures like proteins and hypergraphs.

**Questions:**

1. Can you provide more intuition or theoretical analysis for why Fock space concepts should be particularly well-suited for graph encoding?

2. What would be the performance if FoGE were replaced with strong modern GNNs in the same LLM framework? Given that many advanced GNNs actually have relatively small parameter counts, the "parameter-free" advantage might not be as significant as claimed. This comparison would help isolate whether the benefits come from the Fock space encoding itself or simply from having a structured graph representation.

**Ethical Concerns:**

["NO or VERY MINOR ethics concerns only"]

**Final Justification:**

I think most of my concerns have been addressed; the only concern is that although the evaluation may seem expensive, it is not very comprehensive for both the LLM and GNN communities, such as only considering LLaMA2-7B and evaluating on a limited graph benchmark dataset. However, the innovation of this work is commendable, so I evaluate it as a borderline accepted manuscript and respect the views of other reviewers and AC.

**Limitations:**

See Weakness

**Paper Formatting Concerns:**

None identified

**Quality:**

3

**Strengths And Weaknesses:**

**Strengths**

1. I appreciate the bold attempt to connect quantum mechanics with graph learning. It's refreshing to see researchers exploring mathematical frameworks from physics, even if the connection isn't immediately obvious.

2. The method avoids the complexity of training additional parameters, which could be computationally efficient and reduce overfitting concerns compared to traditional GNN-based approaches.

3. The authors show their method works across quite diverse graph types, which suggests it might be more general than domain-specific approaches.

**Weaknesses**:

1. I'm having trouble seeing why Fock space, which was designed for quantum particle systems, should naturally fit graph structures. The motivation seems to be "we have this cool mathematical tool from physics, let's see if it works" rather than "here's a specific problem with current methods that quantum concepts can solve." This makes the whole approach feel somewhat unmotivated.

2. While the paper introduces an intriguing approach, it would benefit from deeper theoretical examination. The authors claim the encoding provides rich and informative representations, but more analysis would help readers understand what specific graph properties are captured or preserved. Questions like how this encoding theoretically compares to existing methods, what makes it potentially advantageous, and under what conditions it should work well could strengthen the theoretical foundation significantly.

3. The paper would benefit from a more systematic evaluation approach. Currently, the authors jump directly to LLM applications without first establishing how well the Fock space encoding works for graph representation learning itself.

---

> ### Author Rebuttal · Authors · 2025-07-30
>
> We sincerely thank the reviewer for their time and effort in evaluating our work. Below, we address each of their concerns in detail. We hope our responses clarify any reservations and contribute to a stronger support of our paper.
>
> ## Weakness/Question 1: Fit of Fock Space to graphs
>
> If we view the title literally, this paper applies Fock space for graph encoding, which could be  intellectually interesting but indeed, in this case, your comment would be partly valid. But based on the explanation below, you will likely agree that Fock spaces are not our starting point: they come up as an important design choice in a sequence of modeling steps.
>
> Consider a graph that we want to encode. It is common to take the graph Laplacian to give the graph’s structural properties. Its square root (discrete Dirac operator) follows immediately from spectral graph theory. Now, Dirac operators naturally generate Clifford algebra representations - this observation has been used in several recent papers that apply Clifford algebras for graph-like structures, albeit for modeling dynamical systems or message passing on topological structures. The graph to Clifford link in existing works is nice, but for graph encoding specifically, we know that using Clifford algebra directly does not yield radical benefits due to the 2^n dimensional complexity. So, we are partially stuck.
>
> This is exactly where Fock spaces enter the story. As we show in Section 2.2 (lines 145-161), invoking Fock spaces via the spinor representation reduces this 2^n complexity while providing a very useful insight: creation and annihilation operators. Why is this important? Given the already known connections (although from much older literature) between Fock spaces and Vector Symbolic Architectures, we can then use very effective practical options for creation/annihilation: this step makes VSA use theoretically grounded, and allows a unified treatment of different graph types with identical machinery and strong performance which all reviewers appreciated. We hope this answer was satisfactory, but please let us know and we will be very happy to engage more on this point!
>
> ## Weakness 2: Deeper theoretical examination
>
> We appreciate the suggestion to more clearly write down the theoretical benefits.
>
> FoGE preserves several key graph properties that existing methods often sacrifice:
>
> a) Unlike spectral or neural methods that often compress graph information, our encoding is mathematically lossless (lines 217-224). Given an embedding, we can recover any edge $(i,j)$ by evaluating $p_j^T(p_i^{-1} \otimes g)$, and graph size via $p_i^T(s^{-1} \otimes g)$. This property is rare in graph encoding literature to our knowledge.
>
> b) The creation/annihilation operator framework, while instantiated heuristically, naturally handles diverse graph components (nodes, edges, hyperedges, attributes) through the same algebraic operations like $\otimes$ and $\oplus$. Most methods will require architectural changes for different graph types, but we can use the same machinery.
>
> We can also enumerate theoretical advantages over some alternatives. Versus spectral methods, we preserve local structure, not just global connectivity patterns. Versus GNNs, there is no over-smoothing, nor architectural constraints on graph size or type. Versus graph transformers: we get linear complexity instead of quadratic attention.
>
> By design, FoGE will work well when: (1) preserving complete structural information is important, (2) we want to handle diverse graph types easily, (3) computational efficiency is desirable. It may be less optimal when task-specific inductive biases are available and more important for the use case. In principle, we can encode multi-scale relationships through the hierarchy: single nodes (grade-1), pairwise interactions (grade-2), higher-order patterns (grade-k). This gives a principled and convenient multi-resolution view.
>
> ## Weakness 3: Evaluation approach- *the authors jump directly to LLM applications without first establishing how well the Fock space encoding works for graph representation learning itself.*
>
> One of our biggest goals/concerns when writing the paper was to provide a complete evaluation of FoGE in isolation and not merely as a module living inside an LLM. So, our experiments included a thorough examination of the encoding capabilities of FoGE, its resilience to noise, the effect of vectors’ dimension, and how well it preserves graph relationships. Below we list what we have shown.
>
> 1. Tables 1 and 2 demonstrate that FoGE is one of the best approaches for biological datasets, such as BioGRID, HumanNet, and PPI.
> 2. Appendix D demonstrates how well our operations function as we increase the size of the vectors and the size of the underlying graphs.
> 3. Appendix E is dedicated to both simple and advanced graph understanding tasks, using 4 different datasets (GraphQA, Jaffe, HyperGraphQA, and mol-HIV). Our results show that FoGE is not lacking in performance in any case, and it is one of the best solutions for real-world datasets like mol-HIV.
> 4. Appendix F shows how well FoGE encodings reflect the real graph relationships, by showcasing that these encodings preserve the clade information of the proteins, although no such information is available during encoding.
> 5. Appendix G generalizes the results of the main paper (Table 1), showcasing FoGE’s capabilities across a collection of 18 different datasets which are part of the OBN Benchmark.
> 6. Finally, Appendix H depicts the significance of the vector dimension across the tasks of GraphQA.
>
> These experiments, we hope you agree, are quite extensive so we presented a summary of all the results in Section 4.1, while the details are pushed to the appendix. We would appreciate any further suggestions that will help us strengthen the results further and showcase even better the capabilities of FoGE.
>
> ## Question 2: FoGE-LLM vs GNN-LLM
>
> The differences between a GNN-based LLM and a FoGE-based LLM are indeed studied in the paper. GraphLLM uses the exact same backbone and overall training procedure as we do, making the comparison as fair as possible. Despite FoGE-LLM being problem-agnostic and with about 25% of GraphLLM’s trainable parameters, we observe no performance degradation at all (Table 4). Additionally, FoGE-LLM is applicable to scenarios where GNN-based LLMs fails, such as hypergraph understanding (Table 5).
>
> Finally, we fully agree that many modern GNNs have a relatively small parameter count. We examine the performance of such models (e.g., GIN, GCN, GAT) and compare it against FoGE on multiple different settings and datasets (Tables 1, 2 of the main paper, Tables 3, 5 of the Appendix). Our findings consistently show that FoGE offers multiple advantages over such models, beyond just a strict reduction in parameter count.
>
> We welcome any suggestions on additional models that the reviewer feels should be part of our experiments. FoGE works quite well and we are confident that any additional experiments will be consistent with what is included in the paper.

---

> ### Comment · Reviewer_nube · 2025-08-03
>
> Thank you for the authors' thoughtful response. I appreciate the clarification on the motivation for using Fock spaces, which does provide a clearer theoretical pathway.
>
> I still have concerns about the evaluation approach (Weakness 3), though I recognize the extensive experiments provided in the paper. The main issue is that the evaluation includes both LLM applications and standard graph tasks, but neither track comprehensively establishes FoGE's quality. The graph learning experiments are limited in scope, and the LLM evaluation only uses a single model. When combining graph encodings with LLMs, performance can be influenced by several factors:
>
>
> 1. **The combinations between prompts and LLMs is known to significantly affect performance.** Without ablation studies isolating these factors, it's impossible to determine whether the good performance comes from FoGE's superior graph encoding or from a fortunate match with Llama2 (7B)'s specific preferences.
>
>
> 2. In the current pre-alignment era of graph-language integration, parameter-free encodings might inherently be more "LLM-friendly" regardless of their actual graph representation quality. Unlike vision-language models where proper alignment training (e.g., CLIP) has been established, graph-language integration lacks such mature methods. Even carefully designed text templates for encoding graphs might achieve strong performance, suggesting that LLMs may be doing most of the heavy lifting rather than the graph encoding itself.
>
>
> **The fundamental question remains: Is FoGE a good graph encoder in general, or is it merely a good encoder for LLMs?** This distinction is crucial for understanding the contribution and the generalizability of the approach. Specifically, the LLM community would want to know whether this is truly a robust prompting approach that works across different LLMs, while the GNN community would want evidence that FoGE genuinely produces superior graph representations. Currently, both questions would benefit from more comprehensive evaluation.
>
> Despite these concerns, I appreciate the novelty of the Fock space approach and its practical value for graph-LLM integration. While additional experiments (e.g., comparison with recent GNN baselines like GPS or Graphormer, or testing across multiple LLMs) would be beneficial, I recognize that the current work makes a valuable contribution to this emerging area.
>
>
> Given the authors' clarifications and the overall contributions, I am willing to raise my assessment.

---

> > ### Author Response · Authors · 2025-08-05
> >
> > We thank the reviewer for their thoughtful engagement and for their willingness to reconsider their assessment. We appreciate the constructive dialogue and would like to offer a few clarifications regarding the remaining points:
> >
> > 1. On the LLM backbone choice: We selected LLaMA2-7B to allow for a fair comparison with GraphLLM, which uses the same backbone. This helps isolate the effect of the graph encoding itself from variations in language model architecture. While repeating the full set of experiments with multiple LLM backbones will be very expensive, we are happy to include experiments with an additional LLM if this is considered essential. We welcome suggestions.
> >
> > 2. On the generality of FoGE: The question of whether FoGE is a good encoder for graphs in general or specifically in the context of LLMs is important. Our standalone experiments (Tables 1–2, Appendices) address this by evaluating FoGE on classical graph learning tasks, without any LLM involvement. The strong results on PPI and various OBNB datasets suggest that FoGE performs well as a general-purpose graph encoder. We are happy to elaborate further in the paper if the reviewer believes it would be beneficial.
> >
> >  3. On prompting versus encoding contributions: We agree that the interplay between prompts and LLMs can significantly affect performance. Text-based templates of graphs have already been tested in existing works, and the baselines we consider here have already demonstrated that a graph encoder combined with an LLM provides superior performance, similar to a VLM, as the reviewer mentions. We appreciate the suggestion, and we will include these results too for clarity and completeness. Additionally, our ICL experiments (Table 5) on hypergraphs demonstrate a huge performance gap compared to text-based approaches (both zero and few-shot).
> >
> > 4. On contemporary baselines: Appendix E.2 compares FoGE to several strong, engineered models on a publicly available dataset and leaderboard. While our method is unsupervised and parameter-free, it remains competitive with supervised approaches.
> >
> > Once again, we thank the reviewer for their feedback and positive assessment. We believe these clarifications help strengthen the paper and address the main points of concern.

---

### Official Review · Reviewer_aZCB · 2025-07-03

**Clarity:** 2
**Significance:** 3
**Originality:** 3
**Rating:** 5
**Confidence:** 3

**Summary:**

This paper introduces FoGE, a parameter-free graph representation framework inspired by Fock space in quantum mechanics. FoGE encodes graphs as unordered multisets of node features and constructs graph embeddings through a series of multi-linear operations. The method is general-purpose and can handle various graph structures, including simple graphs, hypergraphs, and attributed graphs. Furthermore, FoGE can be combined with a frozen language model via prefix-tuning using a lightweight linear adapter.

**Questions:**

See above.

**Ethical Concerns:**

["NO or VERY MINOR ethics concerns only"]

**Limitations:**

Yes.

**Quality:**

3

**Strengths And Weaknesses:**

Strength:
- The paper introduces a theoretically grounded and lossless graph representation framework inspired by Fock space. It is a convenient method that does not require trainable parameters in the embedding stage.
- This method can generate graph embeddings that are not tailored to any specific downstream task. This suggests strong generalization across tasks and domains.

Weakness:
- The role of the large language model (LLM) in the overall framework is not well explained. It is unclear whether the LLM meaningfully contributes to understanding the graph structure or primarily serves as a text generation component. Further exploration of the interaction between the graph embeddings and the LLM would help clarify its impact and value in the architecture.

---

> ### Author Rebuttal · Authors · 2025-07-30
>
> We thank the reviewer for appreciating our work and its multiple strengths. Below, we address their main concern regarding LLMs.
>
> ## Weakness: Role of large language model (LLM) in the overall framework is not well explained. Unclear whether the LLM meaningfully contributes to understanding the graph structure or primarily serves as a text generation component.
>
> We appreciate this question about the LLM's role. Briefly, we can read this as: why use an LLM at all and is it doing much? Our initial experiments (Section 4.1) show simple models can process FoGE embeddings quite well for traditional tasks. The rationale for LLMs was due to two advantages: (1)  language interface for flexible graph queries without pre-defining task types, and (2) easier integration with mature software ecosystems built around LLMs that reduce deployment overhead.
>
> So does the LLM understand graph structure? Our experimental evidence suggests yes: the model does learn to reason about graph properties. For example, consider compositional reasoning: On complex tasks like bipartite matching, the LLM must understand by processing the encodings how local properties compose into global structures. Or for structural tasks, like max triplet sum or shortest paths, the LLM can also answer such questions successfully, indicating structural understanding.
>
> While this still needs more development, it mirrors multimodal success: just as LLaVA learns visual concepts and TimeLLM develops temporal reasoning once modalities are properly embedded, our results suggest LLMs can develop graph reasoning abilities when provided with rich structural encodings (like FoGE).

---

> > ### Comment · Reviewer_aZCB · 2025-08-08
> >
> > Thanks for the further clarification and the examples. My concerns have been well addressed.

---

### Official Review · Reviewer_gsTE · 2025-07-03

**Clarity:** 2
**Significance:** 2
**Originality:** 2
**Rating:** 4
**Confidence:** 3

**Summary:**

This paper proposes a graph encoding method for Large Language Models (LLMs) inspired by Fock Spaces, a concept from mathematical physics, with the goal of designing generalizable task-agnostic graph embeddings that can be adapted into LLM embedding space for graph-based tasks. Based on concepts from Clifford Algebras and Fock Spaces, a graph embedding is obtained for a given input graph by initializing a vector space for node embeddings and summing up circular convolutions of node embedding pairs based on the edges present in the graph. This creates a graph embedding which preserves the structure of the graph, which then is transformed by a learned linear layer into the embedding that is used by the LLM to represent the graph.

**Questions:**

Questions:
For the basic graph understanding tasks (Table 2, Section E.1) in the Appendix, it is unclear whether the performance on tasks is accuracy or MSE, making it difficult to gauge performance. For example, on GraphQA and HyperGraphQA, it is unclear for the num edges and num nodes task whether the performance indicates MSE or accuracy.
The authors reference results on nineteen real protein datasets from PPI and OBNB in Tables 1 and 2 in Results section 4.1, however it is unclear in the Table captions whether the performance is reported over nineteen datasets or rather the ones referenced in each caption. Clarifying the datasets in each Table and caption would help readability of the results.
Can the authors provide analysis or discussion about the scalability of the method to larger graphs? For instance, on larger graphs of over 1 million nodes, how long would computing the graph embedding take? Additionally, would the dimensionality of the vector become a bottleneck in the ability to encode larger graphs?

**Ethical Concerns:**

["NO or VERY MINOR ethics concerns only"]

**Final Justification:**

FoGE is a novel, parameter-efficient graph encoding method inspired by Fock spaces, applicable across diverse graph types including hypergraphs. Strengths include originality, scalability, and theoretical grounding. My earlier concerns on baseline coverage, statistical reporting, and evaluation breadth were largely addressed in the rebuttal. Some limitations remain, but the contribution is significant and timely, warranting a borderline accept.

**Limitations:**

The method lacks comprehensive comparisons to existing baselines in the GNN + LLM space. However, with more consistent baseline evaluation and more analysis of parameter efficiency versus baseline methods, the quality of the submission could be increased.

**Quality:**

2

**Strengths And Weaknesses:**

Strengths
Parameter Efficiency: The proposed method is parameter-free except for learned linear adapters that transform the graph embedding space into the LLM embedding space. The LLM is frozen in the experiments, giving the proposed method an advantage in terms of parameter-efficiency.
Originality: The method is novel in its formulation of graph encoding based on concepts from Fock space representations, and presents a new way of encoding graphs for LLMs for diverse graph domains. Additionally, the method allows for the encoding of node attributes, a limitation of some previous Graph Neural Network (GNN) + LLM works. The method is generalized up to hypergraphs as well, improving its applicability to various graph domains.
Evaluation Datasets: The authors benchmark their method on six graph datasets, including graph reasoning datasets (GraphQA) and protein graphs (OBNB, PPI). This highlights the applicability of the proposed method.

Weaknesses
Performance Versus Challenging Baselines: FoGE demonstrates similar, sometimes better performance to GNN baselines in the first phase of results (evaluating graph embedding quality) and GNN + LLM baselines in the second phase of results (grounding LLMs with graph prompting). Because the performance of the method is not consistently and significantly higher than baseline methods, the highlight of the method is its novelty and efficiency in graph encoding. This could be emphasized more by adding in trainable parameter counts in more of the results, as the authors did in Table 4.
Significance Values: Standard error in performance is not reported in any of the results tables, except for Table 3 in the appendix. Reporting standard error in more results would increase confidence in the performance gains FoGE does see over baseline methods (e.g. in Table 3 and 5 in the main text).
Baseline Methods & Statistical Significance: Performance results versus other GNN + LLM works (Tables 3 through 5 in the main text) compare against different baseline models in each comparison, leaving a complete comparison on each dataset incomplete. Benchmarking against GraphToken, GraphLLM, and other prefix-tuning methods such as regular prompt tuning of the LLM with a textual graph description consistently on each dataset would give a more complete performance comparison.

---

> ### Author Rebuttal · Authors · 2025-07-30
>
> We sincerely thank the reviewer for the time and effort in evaluating our work. Below, we address each of the concerns in detail. We hope our responses clarify any reservations and contribute to a stronger support of our submission.
>
> ## Weakness 1: Performance Versus Challenging Baselines
>
> We are delighted that the reviewer recognizes the multiple strengths of FoGE, beyond just the raw performance measures. The unsupervised nature of FoGE, its training-free approach, and the limited amount of trainable parameters in the LLM-related experiments are indeed some of the unique qualitative advantages of FoGE over GNN-based methods.
>
> We much appreciate the reviewer’s suggestion, and we will include more information, similar to Table 4, about the adapters size, the number of graph embeddings, as well as other qualities of each approach for each experiment, to further highlight the strengths of FoGE.
>
> ## Weakness 2: Significance Values
>
> We appreciate the suggestion. Significance values were included for experiments where such information was available for all baselines or could be reasonably estimated, without a very large increase in the compute budget required to re-run the experiments. Based on this comment, we will include error bars/intervals for our method for all experiments.
>
> ## Weakness 3: Baseline Methods (Performance results versus other GNN + LLM works on all benchmarks incomplete)
>
> We thank the reviewer for this comment. You will see from the explanation below that our comparisons may appear incomplete because the baselines cannot adapt to all the different types of datasets we have used without very extensive changes. This is one of the main strengths of FoGE, which we highlight with these experiments, but we agree that it may give the wrong impression that method X was intentionally not run on dataset Y.
>
> - Neither GraphLLM nor GraphToken are able to handle hypergraphs; so, in Table 5, we compare only with text-based approaches: zero and few-shot learning, as also noted in the review.
> - Similarly, GraphToken is not able to handle the datasets proposed by GraphLLM in Table 4, hence no comparison with GraphToken is meaningful in that case. We mention this a few times in the paper (e.g., lines 354, 389), but if suggested, we are happy to emphasize further.
> - Finally, comparison with text-based approaches of GraphToken and GraphLLM has already been established in those papers. Based on this suggestion, we will add these results too in the appendix.
>
> We appreciate any suggestions or recommendations of additional baselines the reviewer believes are important to include.
>
> ## Question 1: Performance metrics on Table 2 of Appendix
>
> All results except the “has_cycle” are measured using MSE, since they correspond to regression tasks. The “has_cycle” task corresponds to accuracy (we used % sign but will say accuracy clearly). We apologize for any confusion. We will update the text so that the measurement details are more prominent.
>
> ## Question 2: Results on PPI and OBNB
>
> The complete set of results on the 18 datasets of OBNB are included in the appendix (in Table 5). There we can check the performance of FoGE on all datasets, compared with multiple strong supervised and unsupervised baselines, such as GCN, GAT, and GIN. Table 2 in the main paper demonstrates FoGE’s performance on PPI, showcasing that, although our setup is unsupervised, it is superior to most supervised methods, with an F1 score of 99.2. Finally, Table 1 included only a subset of the results on OBNB, due to space restrictions (the complete results are located in Table 5 of the appendix).
>
> We are sorry if this was not clear, and we will update the manuscript to avoid any confusion.
>
> ## Question 3: Scalability discussion and dimensionality dependence
>
> We appreciate this nice suggestion, and we will emphasize it much more. The complexity of the encoding is provided around line 265: it is linear in the size (number of edges) of the graph, and the specific aggregation we employ only has a complexity of $d\log(d)$, where $d$ is the dimension of the vectors. This is among the strongest advantages of FoGE, and we are happy to make this more prominent in the paper.
>
> Below, we show the runtime of FoGE as we increase the number of edges, on a conventional consumer CPU. The graphs are randomly generated instances of Erdos-Renyi graphs.
>
> | number of edges | 1 | 500 | 50000 | 1200000 | 5000000 |
> | --- | --- | --- | --- | --- | --- |
> | **FoGE runtime (sec)** | 0.0006 (0.0003) | 0.041 (0.0074) | 3.769 (0.3573) | 94.041 (1.0427) | 378.018 (4.5523) |
>
> From above, three observations stand out:
>
> 1. The linear relationship between the number of edges and the runtime.
> 2. Fast encoding of graphs with even millions of edges.
> 3. The ability of FoGE to handle huge graphs without any substantial increase in memory consumption.
>
> Additionally, given the properties of the aggregation we use, the operations can be parallelized on multiple CPU threads (or GPU), speeding up even further the computation on larger graphs. For example, similarly to our experiments with OBNB, we can calculate in parallel a single embedding per node, and then aggregate them together. This can lead to significant improvement in encoding, exploiting the fact that our aggregation operations are lightweight and can even run on CPUs.
>
> Regarding the dimensionality becoming a bottleneck part of the question, we agree that the vector dimensionality can be a factor that can affect performance (and this is true for any encoder). For our construction specifically, we provided detailed studies of the vectors’ dimension impact on the quality of the results (Appendix D and H). Additionally, much more details on the generic encoding capabilities of the operations used are also discussed in [43, 44] in a different context.

---

> > ### Comment · Reviewer_gsTE · 2025-08-06
> >
> > Thank you for your detailed clarifications and response. I acknowledge your points and updates and will take them into account in my assessment.

---

> > ### Comment · Reviewer_gsTE · 2025-08-08
> >
> > Thank you for clarifying the runtime and complexity aspects in your rebuttal. One aspect I am still curious about is the quality of FoGE’s representations at extreme scales. Have you investigated whether the preservation of structural information, especially for higher-order patterns or long-range dependences, changes when moving to graphs much larger or denser than those in your current experiments (e.g beyond the OBNB datasets)?  Relatedly, does maintaining representational fidelity in such settings require increasing the vector dimensionality, and if so, how does that trade off with the method’s efficiency?

---

> > > ### Author Response · Authors · 2025-08-08
> > >
> > > We thank the reviewer for the question. Yes, it is interesting to study representation fidelity at scale. This touches upon some fundamental limits that any graph encoding method must confront to, which we describe below.
> > >
> > > For a graph with $V$ nodes and $E$ edges, the full structural information requires information of the order $E \log(V)$, since each edge must be specified by its two nodes, and $V$ nodes require about $\log(V)$ bits of information. Compressing this into a fixed representation of dimensionality $d$ must involve information loss in the case that $d$ is (much) smaller. This is not a limitation of FoGE: no method can losslessly encode arbitrarily large graphs into a fixed-size representation without eventual fidelity degradation.
> > >
> > > In VSA frameworks (which drive our practical implementation), the recovery accuracy for structural queries degrades as $\mathcal{O}\big(\frac{1}{\sqrt{d}}\big)$, where $d$ is the vector dimensionality. For graphs with millions of edges, maintaining high fidelity would indeed require an impractically large dimension $d$, which we think is the intuition behind your question. For fixed $d$, as we "squeeze" more nodes and edges, the SNR starts to drop, which impacts recovery.
> > >
> > > One approach is of course to increase $d$, so that it matches the theoretical rule of thumb. However, in practice, rather than forcing a single global vector to capture a massively large graph, we can transition to node-level encodings, just like all GNN-based architectures do and as we demonstrate in the OBNB experiments that consist of thousands of nodes. This maintains the theoretical advantage and, in this case, FoGE's parameter-free node-level embeddings match or exceed heavily parameterized methods such as GIN, GAT, and GCN. This confirms that our setup can gracefully handle the scale-fidelity trade-off.

---

### Official Review · Reviewer_dfsh · 2025-07-05

**Clarity:** 3
**Significance:** 3
**Originality:** 3
**Rating:** 5
**Confidence:** 2

**Summary:**

The paper presents a novel parameter-free Fock space based-graph encoder to express graph-structured data in diverse applications. Existing works generally train graph Transformers or simply serialize the graph with triplets or paths. The proposed Fock space-inspired representation effectively encodes graph. The authors’s experiments, the proposed method demonstrates the effectiveness of the proposed method.

**Questions:**

- Could you provide the results on large-scale graphs?
- Could you provide additional experimental results based on other large language models?
- I think that the proposed method is also applicable to prompt tuning instead of prefix tuning. Could you have any experimental results based on prompt tuning?

**Ethical Concerns:**

["NO or VERY MINOR ethics concerns only"]

**Final Justification:**

I have read all the reviews and the corresponding responses. Most of my concerns have been addressed and I decided to maintain my score.

**Limitations:**

The authors have adequately addressed the limitations.

**Paper Formatting Concerns:**

There is no concern.

**Quality:**

3

**Strengths And Weaknesses:**

## Strengths

- The proposed method is clearly written to understand the paper.
- The proposed method seems novel to me.
- The paper demonstrates the effectiveness of the work theoretically and empirically.

## Weaknesses

- From my thoughts, the proposed method has a limitation un handling a large scale graph because it has to express both edges and vertexes. Could you discuss it?
- It would be better if the authors included more results based on diverse large language models such as Qwen or Gemma.

---

> ### Author Rebuttal · Authors · 2025-07-30
>
> We thank the reviewer for their time and effort. Below we address all of the concerns brought up in the review.
>
> ## Question 1: Provide the results on large-scale graphs?
>
> Thanks for the suggestion. We indeed experimented with large-scale graphs, this was described in Section 4.1. For example, OBNB consists of graphs with tens of thousands of nodes (e.g., BioGRID has 18951 nodes) and our implementation currently scales quite well. In these cases and to be consistent with existing works, we produce a single encoding per node (mentioned around line 323). We find that our encodings are a strong alternative and perform even better than trainable methods in many cases! We are happy to expand this text more.
>
> The reviewer will also like that the complexity of the encoding (described around line 265), is *linear* in the size (number of edges) of the graph. The specific aggregation we use has a complexity of only $d\log(d)$, where $d$ is the dimension of the vectors. So scalability is really a strong benefit.
>
> To fully address this concern, we also show the runtime of FoGE below as we increase the number of edges, on a conventional consumer CPU. The graphs are randomly generated instances of Erdos-Renyi graphs.
>
> | number of edges | 1 | 500 | 50000 | 1200000 | 5000000 |
> | --- | --- | --- | --- | --- | --- |
> | **FoGE runtime (sec)** | 0.0006 (0.0003) | 0.041 (0.0074) | 3.769 (0.3573) | 94.041 (1.0427) | 378.018 (4.5523) |
>
> From above discussion, three points are clear:
>
> 1. Linear relationship between the number of edges and the runtime.
> 2. Fast encoding of graphs with even millions of edges.
> 3. The ability of FoGE to handle huge graphs without any large increase in memory consumption.
>
> Additionally, given the properties of the aggregation we use, the operations can be easily parallelized on multiple CPU threads, speeding up the implementation even further on larger graphs. For example, similar to our experiments with OBNB, we can calculate in parallel a single embedding per node, and then aggregate them together. This can lead to significant improvement in encoding, since our aggregation operations are lightweight and can even run on CPUs.
>
> ## Question 2: Additional results on other LLMs
>
> Our approach is backbone-agnostic by design: the graph encodings are converted to embeddings via simple linear adapters that can interface with any LLM. We chose LlaMA since it is widely used and was also the LLM backbone in one of our baselines (GraphLLM). While repeating the full set of experiments with multiple LLM backbones will be very expensive, we are happy to include experiments with one additional LLM if this is considered essential. We welcome suggestions.
>
> ## Question 3: Prefix tuning vs prompt tuning
>
> We appreciate this question, but we believe there may be some confusion about terminology. Prompt tuning typically refers to learning soft prompt tokens in the embedding space, while discrete prompting involves crafting text-based prompts. Our graph encodings are high-dimensional continuous representations that is converted to the LLM's embedding space via linear adapters - this is different from prompt tuning approaches that optimize token-level embeddings. The graph structural information we encode will be difficult to be meaningfully expressed through discrete text prompts or soft prompt tokens. We like the suggestion, and will include a discussion on prompt tuning versus prefix tuning in our work and include any specific experiment along this direction which will add value.

---

> > ### Comment · Reviewer_dfsh · 2025-08-07
> >
> > Thank you for the authors' thoughtful response. Most my concerns have been addressed. After carefully reading other reviews and the corresponding rebuttals, I keep my score.

---

> > > ### Author Response · Authors · 2025-08-07
> > >
> > > many thanks for asking these questions and for your positive vote of confidence for our work. We much appreciate it. We will make space to incorporate this discussion into the paper. Thanks.

---

### Author Response · Authors · 2025-08-07

Dear reviewers,

We much appreciate your time and effort. We thank you for your engagement and your support to our work. As the discussion period comes to a close, we remain at your disposal for any further clarifications or questions.

Sincerely,
the authors

---

### Note · Authors · 2025-08-12

Dear AC,

We are thankful to all reviewers for the high-quality reviews. The discussion period was fruitful, and we appreciate the insightful questions. All reviewers liked our work, and we are certain that these discussions will improve the quality of our final version even more. There was a minor question towards the end of the discussion, but we hope that our response was sufficient to reassure the reviewer. Again, many thanks for your time.

---

### Decision · Program_Chairs · 2025-09-17

**Decision:**

Accept (poster)

**Comment:**

This paper addresses the problem of designing a graph embedding to be used as a prompt to a pretrained LLM. The paper proposes a graph encoding method inspired by Fock Spaces, a concept from mathematical physics, with the goal of designing generalizable task-agnostic graph embeddings that can be adapted into LLM embedding space for graph-based tasks. Based on concepts from Clifford Algebras and Fock Spaces, a graph embedding is obtained for a given input graph by initializing a vector space for node embeddings and summing up circular convolutions of node embedding pairs based on the edges present in the graph. This creates a predefined graph embedding (with no trainable parameters) which mostly preserves the graph information. This embedding is then transformed by a learned linear layer to define the embedding that is used as an input to the LLM. The LLM here is pretrained and frozen, namely, in training, only the linear layer is optimized.

The reviewers were mostly positive about this submission, stating that it is timely, the graph embedding is novel, efficient, theoretically grounded, and applicable across diverse types of graphs.